# Association of genetic variations in *ACE2*, *TIRAP* and *factor X* with outcomes in COVID-19

Marissa J. M. Traets[1]☯*, Roel H. T. Nijhuis[2], Servaas A. Morré[3,4], Sander Ouburg[3], Jasper A. Remijn[5], Bastiaan A. Blok[1], Bas de Laat[6], Eefje Jong[1], Gerarda J. M. Herder[7], Aernoud T. L. Fiolet[1]☯, Stephan P. Verweij[1,8]☯

**1** Meander Medical Centre, Department of Internal Medicine, Amersfoort, The Netherlands, **2** Meander Medical Centre, Department of Medical Microbiology and Medical Immunology, Amersfoort, The Netherlands, **3** Department of Medical Microbiology and Infection Control, Laboratory of Immunogenetics, Amsterdam UMC, Amsterdam, The Netherlands, **4** Department of Genetics and Cell Biology, Institute for Public Health Genomics, Research Institute GROW, Faculty of Health, Medicine & Life Sciences, University of Maastricht, Maastricht, The Netherlands, **5** Meander Medical Centre, Department of Clinical Chemistry, Amersfoort, The Netherlands, **6** Synapse Research Institute, Maastricht, The Netherlands, **7** Meander Medical Centre, Department of Pulmonary Disease, Amersfoort, The Netherlands, **8** Department of Respiratory Medicine, University Medical Centre Utrecht, Utrecht, The Netherlands

☯ These authors contributed equally to this work.
* mjmtraets@gmail.com

**Data Availability Statement:** All relevant data are within the manuscript.

**Funding:** The author(s) received no specific funding for this work.

## Abstract

### Background

Coronavirus disease 2019 (COVID-19), caused by the severe acute respiratory syndrome coronavirus 2 (SARS-CoV-2), can manifest with varying disease severity and mortality. Genetic predisposition influences the clinical course of infectious diseases. We investigated whether genetic polymorphisms in candidate genes *ACE2*, *TIRAP*, and *factor X* are associated with clinical outcomes in COVID-19.

### Methods

We conducted a single-centre retrospective cohort study. All patients who visited the emergency department with SARS-CoV-2 infection proven by polymerase chain reaction were included. Single nucleotide polymorphisms in *ACE2* (rs2285666), *TIRAP* (rs8177374) and *factor X* (rs3211783) were assessed. The outcomes were mortality, respiratory failure and venous thromboembolism. Respiratory failure was defined as the necessity of >5 litres/minute oxygen, high flow nasal oxygen suppletion or mechanical ventilation.

### Results

Between March and April 2020, 116 patients (35% female, median age 65 [inter quartile range 55–75] years) were included and treated according to the then applicable guidelines. Sixteen patients (14%) died, 44 patients (38%) had respiratory failure of whom 23 required endotracheal intubation for mechanical ventilation, and 20 patients (17%) developed venous thromboembolism. The percentage of *TIRAP* polymorphism carriers in the survivor group was 28% as compared to 0% in the non-survivor group (p = 0.01, Bonferroni corrected p =

**Competing interests:** The authors have declared that no competing interests exist.

0.02). Genotype distribution of *ACE2* and *factor X* did not differ between survivors and non-survivors.

## Conclusion

This study shows that carriage of *TIRAP* polymorphism rs8177374 could be associated with a significantly lower mortality in COVID-19. This *TIRAP* polymorphism may be an important predictor in the outcome of COVID-19.

## Introduction

The coronavirus disease 2019 (COVID-19) pandemic is caused by the severe acute respiratory syndrome coronavirus 2 (SARS-CoV-2). Since December 2019, the virus quickly spread around the world and the World Health Organization reported more than 182 million patients as of June 2021 [1]. COVID-19 is associated with a heterogeneous clinical course, varying from asymptomatic to severe disease requiring mechanical ventilation. Various (clinical) factors are associated with a more severe course of COVID-19, among which are male gender, older age, and comorbidities, such as hypertension, cardiovascular disease, diabetes mellitus, chronic lung disease and cancer [1, 2]. Diversity in the clinical course of COVID-19 may be related to differences in virulence among SARS-CoV-2 strains [3]. Genetic predisposition might influence disease course, as is seen in other respiratory viral infections, such as influenza and respiratory syncytial virus [4].

SARS-CoV-2 is member of the Coronaviridae family. The receptor-binding domain of the spike protein of SARS-CoV-2 binds to the angiotensin-converting enzyme 2 (ACE2) receptor which leads to membrane fusion and entry of SARS-CoV-2 into the target cells [5]. Differences in the function of ACE2 might influence susceptibility of COVID-19 [6]. After cellular entry of SARS-CoV-2, viral infection is likely sensed by the immune system through pattern recognition receptors, such as toll-like receptors (TLRs) that serve to recognise pathogens like coronaviruses [7]. Since SARS-CoV-2 is a single-stranded RNA virus, TLR7 seems likely to be implicated in the initial immune response to the virus [8]. TLRs are also able to detect endogenous ligands resulting in an inflammatory response. For example, oxidised phospholipids are agonists for TLR4 and stimulation of TLR4 was associated with acute respiratory distress syndrome in experimental mouse models for acute lung injury, and TLR4 promotes a strong cytokine release by human alveolar macrophages *in vitro* [9, 10]. Phospholipids present in alveolar surfactant are oxidised by neutrophil myeloperoxidase, reported to be at increased levels in COVID-19 patients [11–13]. Therefore, TLR4 and its subsequent downstream pathway may be of critical importance in SARS-CoV-2 induced acute respiratory distress syndrome and subsequent mortality, even though SARS-CoV-2 does not directly bind to TLR4. Upon stimulation, TLRs activate intracellular adaptor proteins and pro-inflammatory pathways are stimulated. TIRAP, also known as Mal, is a toll/interleukin-1 receptor domain-containing adaptor protein for TLR2 and TLR4. After stimulation, TIRAP recruits MyD88 and triggering the MyD88 transduction signalling pathway results in activation of nuclear factor-κB and the subsequent expression of pro-inflammatory cytokines, such as interleukin-1, interleukin-6 and tumour necrosis factor alpha [14]. **Fig 1** depicts a schematic representation of this pathway. Pro-inflammatory cytokines are associated with a worse clinical outcome in COVID-19 [6, 15]. Interleukin-6 and tumour necrosis factor alpha are predictive in COVID-19 survival [16]. The rs8177374 polymorphism of *TIRAP* attenuates the function of TIRAP, leading to reduced

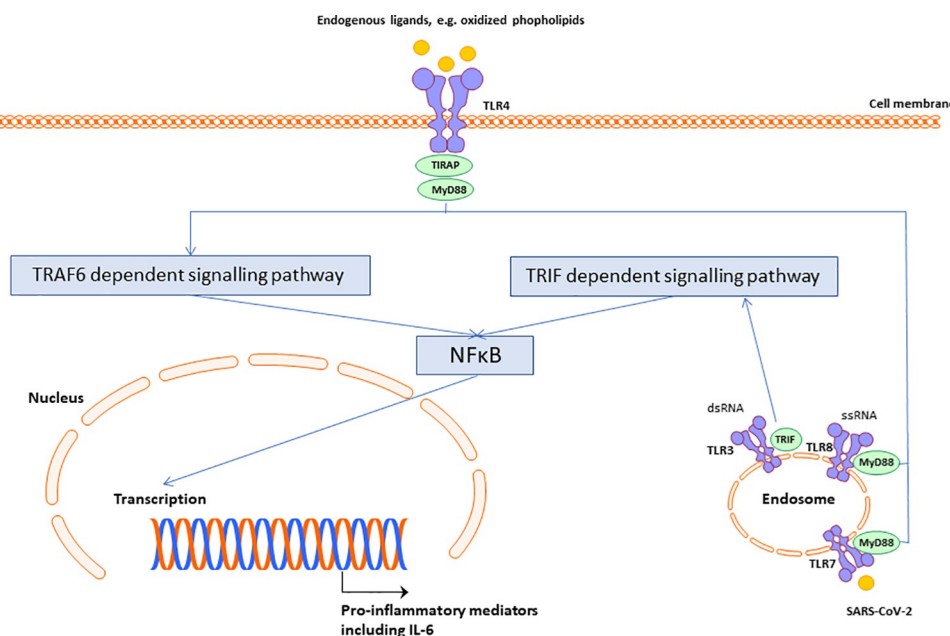

**Fig 1. Proposed simplified schematic representation of SARS-CoV-2 induced intracellular pro-inflammatory signalling pathway.** Abbreviations: IL; interleukin, MyD88; myeloid differentiation factor 88, NFkB; nuclear factor-kappa B, TIRAP; toll/interleukin-1 receptor domain-containing adaptor protein, TLR; toll-like receptor, TRAF6; TNF Receptor Associated Factor 6, TRIF; TIR-domain-containing adapter-inducing interferon-β, ssRNA; single-strand ribonucleic acid, dsRNA; double-strand ribonucleic acid.

production of pro-inflammatory cytokines [17, 18]. This polymorphism has been associated with an attenuated course of several infectious diseases, but has not been studied in relation to SARS-CoV-2 [17, 18].

COVID-19 is often complicated by thrombo-embolic events [19]. Coagulation and inflammation are related, suggesting that coagulation plays a role in COVID-19 [20]. Polymorphisms in coagulation *factor X* may be associated with infection related disorders [21]. A phenome-wide association study revealed that missense mutation rs3211783 was associated with various infections, including viral infections, but not with thrombo-embolic events [21]. Protease factor Xa cleaves the SARS virus S protein into S1 and S2 subunits when the virus enters the target cells, which increases infectivity [22].

Abnormalities in expression of substituents of TLR induced pro-inflammatory signalling pathways, such as *TIRAP*, or abnormalities in *ACE2* or *factor X* may alter course of a SARS-CoV-2 infection. Single nucleotide polymorphisms (SNPs) in these genes may modify expression or affect the function of translated proteins. Variations in the function of these proteins might partially explain the heterogeneous clinical course of COVID-19. Therefore, we hypothesised that carriage of selected SNPs in candidate genes *ACE2* (rs2285666), *TIRAP* (rs8177374), and *factor X* (rs3211783) are associated with the clinical course of SARS-CoV-2 infection.

## Materials and methods

### Study design

The Markers in COVID-19 And Relations to Outcomes in the Netherlands (MACARON) study was a single-centre retrospective cohort study. The study protocol was approved by the local Medical Ethical Committee (TWO 20–043).

## Setting and study population

All patients with a clinical suspicion of SARS-CoV-2 infection who visited the emergency department of the Meander Medical Centre in Amersfoort, the Netherlands between 23 March and 20 April 2020 were enrolled in the study. Blood samples were obtained upon presentation at the emergency department. Adults of Western European descent with proven SARS-CoV-2 infection were eligible for SNP determination. A nasopharyngeal swab was obtained for the detection of SARS-CoV-2 by reverse-transcriptase polymerase chain reaction (PCR). Patients were followed during the entire period of hospitalisation until discharge and treated according to the Dutch guidelines in March and April 2020, which at that time included treatment with chloroquine for patients with respiratory failure [23]. Patients who visited the emergency department but were not hospitalised were assumed to have no respiratory failure or venous thrombo-embolism at time of consultation.

## Data collection

Clinical data were collected using an electronic case report form based on a template of the World Health Organization [24]. Patient characteristics at baseline, symptoms on admission and outcomes were extracted from the electronic medical records. Comorbidities and use of co-medication were collected as present at the time of diagnosis of COVID-19. Pulmonary disease was defined as one of the following diseases: chronic obstructive pulmonary disease, asthma, interstitial lung disease, obstructive sleep apnoea or lung cancer. Auto-immune or inflammatory disease was defined as any rheumatologic disorder, inflammatory bowel disease or sarcoidosis. Malignant neoplasm was defined as a current or past solid organ or haematological malignancy (non-melanoma skin cancers excluded). Antithrombotic drugs were defined as therapy with an anti-platelet agent (acetylsalicylic acid, adenosine diphosphate receptor inhibitors or dipyridamole) or anticoagulants (direct oral anticoagulants or coumarins), or both.

Respiratory failure, venous thromboembolism and mortality were determined as the main outcome measures of the study. Respiratory failure was defined as the necessity of >5 liters/minute oxygen, high flow nasal oxygen suppletion or mechanical ventilation. Venous thromboembolism was defined as pulmonary embolism or deep vein thrombosis diagnosed by computed tomography angiography for pulmonary embolism or ultrasonography for deep vein thrombosis. Mortality was defined as in hospital mortality and within 14 days after palliative discharge.

## DNA isolation and SNP determination

Routine blood tests were performed in the emergency department and plasma remnants were used for laboratory testing. Plasma was separated from blood according to local protocol and was subsequently stored at -70 degrees Celsius prior to DNA extraction. DNA was extracted from 200μl plasma with the fully automated QIASymphony system (Qiagen, Venlo, the Netherlands) using the DSP DNA Mini Kit (Qiagen), yielding 200μl eluate to be used for further processing. SNP databases were used to assess background incidence of the studied SNPs. The following SNPs were assessed and analysed for this study: *ACE2* rs2285666, *TIRAP* rs8177374, *factor X* rs3211783. Carriage included both heterozygote and homozygote mutation carriers. For the *ACE2* and *TIRAP* SNPs, real time PCRs with specific primer pairs and probes (Biolegio Lab Equipment B.V., Nijmegen, the Netherlands) were performed to determine the various alleles, see **Table 1** for primers and probes sequences. Genotyping was performed using the CFX96 system (Bio-Rad Laboratories Inc., California, United States of America) and analysed with Bio-Rad CFX manager 3.1. The total volume per sample was 20 microliter (5 μl DNA,

**Table 1. *ACE2* and *TIRAP* primers and probes sequences.**

| *ACE2* | Forward primer | 5'-ATC TAT GTG TTG AAA CAC ACA TAT CTG C-3' |
|---|---|---|
| | Reverse primer | 5'-AGA TAA TCC ACA AGA ATG CTT ATT ACT TGA-3' |
| | Probe C | 5' 6FAM-CAC TAC TAA AAA TTA GTA GCC TAC-3' |
| | Probe T | 5' VIC-CAC TAC TAA AAA TTA GTA GCT TAC CT-3' |
| *TIRAP* | Forward primer | 5'-CTG CAG GCC CTG ACC G-3' |
| | Reverse primer | 5'-AAT CGG AGC TCA GGT GGG TA-3' |
| | Probe C | 5' 6FAM-CTG CTG TCG GGC CT-3' |
| | Probe T | 5' VIC-CCC TGC TGT TGG GC-3' |

10 μL LightCycler probes master mix 2x (Roche Diagnostics Netherlands B.V., Almere, the Netherlands), 0.18 μl of both primers (stock concentration 100 μM dissolved in TE buffer), 0.27 μl of both probes (stock concentration 15 μM), 4.1 μl water (Roche Diagnostics Netherlands B.V., Almere, the Netherlands). PCR conditions were 2 minutes at 50˚C, 10 minutes at 95˚C, and 45 cycles of 10 seconds at 95˚C and 1 minute at 60˚C. The *factor X* SNP was ordered as Assay-On-Demand from Applied Biosystems (California, United States of America) and was determined according to manufacturers' protocol on the Bio-Rad CFX system.

## Statistical analysis

Continuous variables with non-normal distribution were presented as a median with an interquartile range (IQR). Categorical variables were expressed as counts and percentages.

Univariate analyses were performed with Fisher's exact test to examine the association between SNP carriage and clinical outcomes. The Mann-Whitney U test was performed to compare biochemical parameters between different genotypes. A p-value of $<0.05$ was considered statistically significant. To correct for multiple testing, the Bonferroni correction was applied when appropriate. Analyses were performed using IBM SPSS Statistics for Windows, version 26 (IBM Corp., Armonk, N.Y., USA).

## Results

### Patient characteristics

Of 484 patients suspected of SARS-CoV-2 infection that visited the emergency department, 116 met inclusion criteria and underwent SNP analysis (**Fig 2**). Patient characteristics are presented in **Table 2**. More than half (65%) were male and just over half (58%) had a body-mass index $> 25$ kg/m$^2$. Hypertension was the most common comorbidity (37%). Median duration of symptoms prior to hospital admission was 8 days (IQR 5–12). Ninety patients (78%) were hospitalised of whom 66/90 (73%) were treated with oseltamivir and 43/90 (48%) were treated with chloroquine. Ten patients (11%) received intravenous prednisone. None of the patients received other antiviral or anti-inflammatory agents, such as tocilizumab or dexamethasone. Median length of hospital stay until discharge was 5 days (IQR 3–11 days). Of the 90 hospitalised patients, 28 patients (31%) had advanced directives excluding admission to the ICU for mechanical ventilation, if this situation would occur.

### Outcomes

During the observation period of the study, 16/116 patients (14%) died of whom 13 patients on the ward, two patients on the Intensive Care Unit and one patient at home (two days after discharge). The mortality group was equal between sexes. Median age of patients that died was

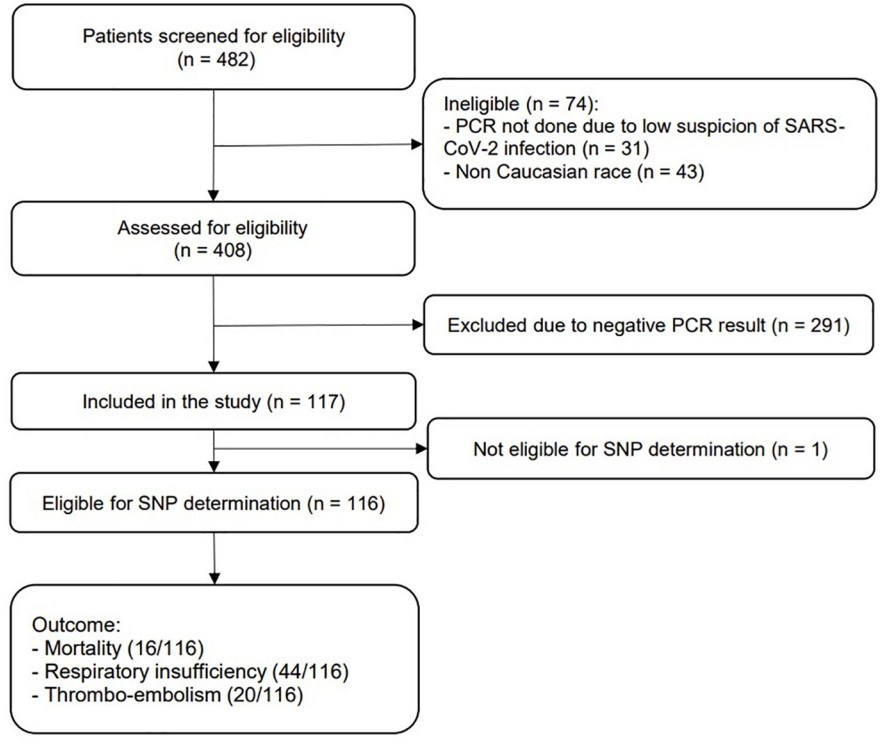

**Fig 2. Flow diagram of patient enrolment.**

75 years (IQR 71–83). Respiratory failure occurred in 44/116 patients (38%) of whom 23 required endotracheal intubation and mechanical ventilation at the intensive care unit. The remaining 21 patients with respiratory failure were treated with oxygen therapy: oxygen mask (n = 5), venturi mask (n = 2), non-rebreathing mask (n = 11) or high flow nasal oxygen suppletion (n = 3). Seventeen percent (20/116 patients) developed a venous thrombo-embolism: 17 patients had a pulmonary embolism, two patients had deep vein thrombosis, and one patient both.

### Genotype distribution

The genotype distribution is shown in **Fig 3**. Determination of the various alleles was successfully in 115 out of 116 patients. Determination of *TIRAP* polymorphism of one patient could not be determined due to technical issues. Genotype distribution for each clinical outcome is shown in **Tables 3–5**. **Fig 4** shows the distribution of the clinical outcomes stratified per genotype. The percentage of *TIRAP* polymorphism carriers in the survivor group was 28% as compared to 0% in the non-survivor group (p = 0.01, Bonferroni corrected p = 0.02). The proportion of carriage of *TIRAP* polymorphism was also markedly higher in patients without respiratory failure, although not statistically significant. No obvious differences in distribution of the *TIRAP* polymorphism and the occurrence of venous thromboembolism were found. No significant differences were observed in genotype distribution for the *ACE2* polymorphism and clinical outcome. None of the patients were carrier of the *factor X* polymorphism. **Table 6** shows levels of C-reactive protein, D-dimer and ferritin in patients with *TIRAP* carriage and *TIRAP* wild-type. Levels of these biochemical parameters show a similar distribution between the two genotype groups at the moment of presentation at the emergency department.

**Table 2. Baseline patient characteristics and outcomes.**

| | | | N = 116 | |
|---|---|---|---|---|
| Median age (IQR)—years | | | 65 | (55–75) |
| Female sex—no. (%) | | | 41 | (35.3) |
| Body-mass index—no. (%)* | | | | |
| | 25–29 kg/m$^2$ | | 43 | (37.1) |
| | $\geq$ 30 kg/m$^2$ | | 24 | (20.7) |
| Comorbidities—no. (%) | | | | |
| | Auto-immune or inflammatory disease | | 14 | (12.1) |
| | Chronic obstructive pulmonary disease | | 16 | (13.8) |
| | Diabetes Mellitus | | 17 | (14.7) |
| | Coronary artery disease | | 19 | (16.4) |
| | Hypertension | | 43 | (37.1) |
| | Cancer | | 19 | (16.4) |
| | History of venous thromboembolism | | 5 | (4.3) |
| | History of ischemic stroke | | 6 | (5.2) |
| Medication at baseline—no. (%) | | | | |
| | Immunosuppressive drugs | | | |
| | | Glucocorticoid | 19 | (16.4) |
| | | Systemic | 7 | (6.0) |
| | | Inhaled | 15 | (12.9) |
| | | Other | 4 | (3.4) |
| | ACE inhibitor or ARB | | 28 | (24.1) |
| | Antithrombotic drugs | | 31 | (26.7) |

*Body-mass index was available for 104/117 patients.

Abbreviations: IQR, inter quartile range; ACE, angiotensin-converting enzyme; ARB, angiotensin II receptor blocker.

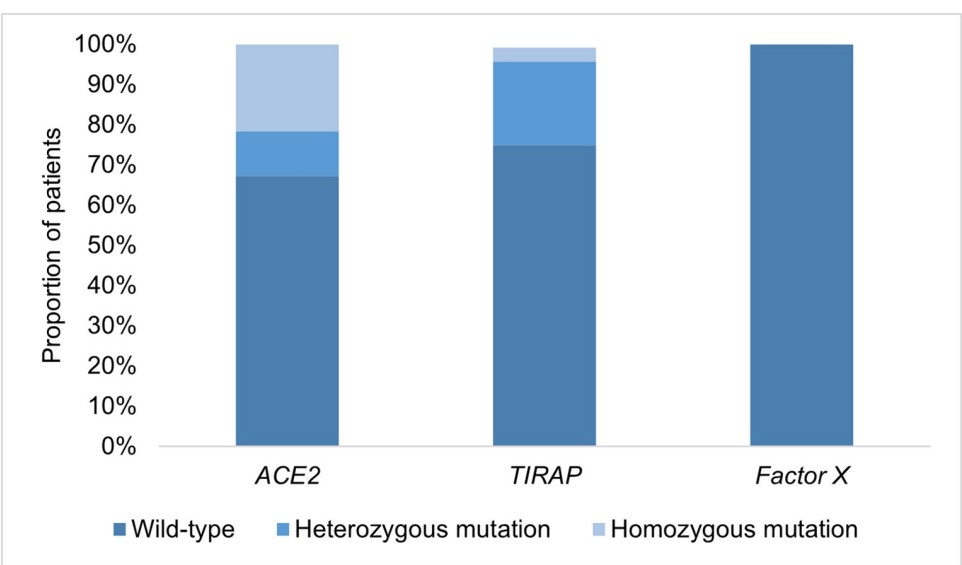

**Fig 3. Genotype distribution.**

**Table 3. Genotype frequencies in survivors and non-survivors.**

| | | Mortality | | | | *P*-value |
|---|---|---|---|---|---|---|
| | | **Yes (N = 16)** | | **No (N = 100)** | | |
| *ACE2* | Carriage | 4 | (25.0%) | 34 | (34.0%) | 0.58 |
| | Wild-type | 12 | (75.0%) | 66 | (66.0%) | |
| *TIRAP* | Carriage | 0 | (0.0%) | 28 | (28.0%) | 0.01 |
| | Wild-type | 16 | (100.0%) | 71 | (71.0%) | |
| *Factor X* | Carriage | 0 | (0.0%) | 0 | (0.0%) | NA |
| | Wild-type | 16 | (100.0%) | 100 | (100.0%) | |

## Discussion

In this study we investigated the potential association between genetic polymorphisms and clinical outcomes in COVID-19. Our findings demonstrated that carriage of the *TIRAP* polymorphism rs8177374 was significantly associated with lower mortality in COVID-19. To our knowledge, no other studies have investigated the association between this polymorphism in the *TIRAP* gene (rs8177374) and clinical outcomes in patients with SARS-CoV-2 infection. This *TIRAP* polymorphism may be associated with a lower risk for respiratory failure, though this finding did not reach statistical significance.

The allele frequency of *TIRAP* polymorphism rs8177374 ranges between 16–34% in the European population which accords to the frequency found in our study (24%) [25, 26]. Increased levels of several pro-inflammatory cytokines and immune hyperactivation are associated with a worse clinical outcome in COVID-19 [6, 15]. One could therefore speculate that decreased cytokine production due to the rs8177374 variant of *TIRAP* leads to a more beneficial clinical course. This *TIRAP* polymorphism could be of value in risk stratification for mortality in hospitalised patients. Furthermore, it may help to identify which patients will benefit from expensive treatment, such as tocilizumab, an anti-interleukin-6 receptor monoclonal antibody. We did not measure levels of cytokines such as interleukin-1 and interleukin-6 in the current study. However, we did measure downstream inflammatory markers such as C-reactive protein, D-dimer and ferritin. C-reactive protein is a surrogate for the interleukin-6 pathway [27]. The results show a similar distribution between patients with *TIRAP* carriage and *TIRAP* wild-type at the moment of presentation at the emergency department. However, this study was not designed to determine an association between the candidate genes and biochemical parameters. The patients in our study presented at different days of onset of SARS-CoV-2 infection and in different clinical conditions at the emergency department. The levels of the biochemical parameters at presentation in the emergency department were expected to vary strongly and therefore cannot be interpreted or be used as functional validation of the

**Table 4. Genotype frequencies in patients with and without respiratory failure.**

| | | Respiratory failure | | | | |
|---|---|---|---|---|---|---|
| | | **Yes (N = 44)** | | **No (N = 72)** | | *P*-value |
| *ACE2* | Carriage | 12 | (27.3%) | 26 | (36.1%) | 0.42 |
| | Wild-type | 32 | (72.7%) | 46 | (63.9%) | |
| *TIRAP* | Carriage | 7 | (15.9%) | 21 | (29.2%) | 0.12 |
| | Wild-type | 37 | (84.1%) | 50 | (69.4%) | |
| *Factor X* | Carriage | 0 | (0.0%) | 0 | (0.0%) | NA |
| | Wild-type | 44 | (100.0%) | 72 | (100.0%) | |

**Table 5. Genotype frequencies in patients with and without venous thromboembolism.**

| | | Venous thromboembolism | | | | P-value |
|---|---|---|---|---|---|---|
| | | Yes (N = 20) | | No (N = 96) | | |
| *ACE2* | Carriage | 4 | (20.0%) | 34 | (35.4%) | 0.20 |
| | Wild-type | 16 | (80.0%) | 62 | (64.6%) | |
| *TIRAP* | Carriage | 4 | (20.0%) | 24 | (25.0%) | 0.78 |
| | Wild-type | 16 | (80.0%) | 71 | (74.0%) | |
| *Factor X* | Carriage | 0 | (0.0%) | 0 | (0.0%) | NA |
| | Wild-type | 20 | (100.0%) | 96 | (100.0%) | |

Data represent the number of patients (%) in each outcome group.

Abbreviation: NA, not applicable.

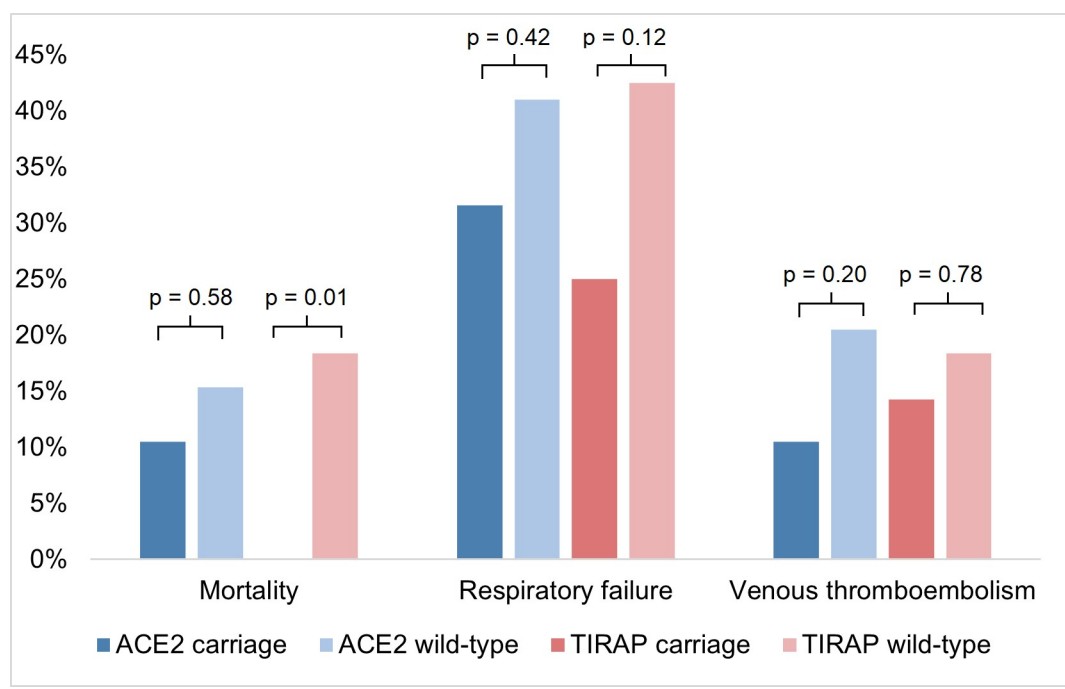

**Fig 4. Clinical outcomes stratified per genotype.**

**Table 6. Biochemical parameters in patients with *TIRAP* carriage and *TIRAP* wild-type.**

| | *TIRAP* | | |
|---|---|---|---|
| | Carriage (N = 28) | Wild-type (N = 87) | P-value |
| C-reactive protein—mg/L | 108 (47–187) | 85 (42–168) | 0.54 |
| D-dimer—mg/L | 1.1 (0.5–2.2) | 1.3 (0.6–2.2) | 0.63 |
| Ferritin - µg/L | 713 (370–1752) | 776 (422–1573) | 0.98 |

Levels are depicted as median and 25th and 75th interquartiles.

*TIRAP* polymorphism. Contrary to biochemical response, the genotypes are not influenced by the day of presentation of the SARS-CoV-2 infection. Consequently, the risk of confounded effect estimates is expected to be low.

*ACE2* polymorphisms, including rs2285666, are associated with differences in gene expression and ACE2 receptor function [28, 29]. One third of our population was carrier of *ACE2* polymorphism (rs2285666), with most patients being homozygous. This proportion was comparable with the allele frequency found in the general European population which is 20–23% [30]. We observed a lower occurrence of mortality, respiratory failure and venous thromboembolism, but whether this is a true signal is unclear as current data are not consistent [28, 29]. We found no carriers of the *factor X* polymorphism, which is concordant with the low allele frequency in the general population [31].

We acknowledge several limitations of the current study. First, the small sample size limits inference of associations in relevant subgroups due to low power. We chose only to include PCR confirmed patients with SARS-CoV-2, excluding any that may have been false negative. Only patients of Western European descent were included as genetic predisposition is different between races. Second, data collection was retrospective with no active follow-up of patients who were not hospitalised. We assumed that patients with clinical symptoms of respiratory failure or venous-thrombo-embolism would be re-admitted. Third, patients were included during the early phase of the COVID-19 pandemic. With advancing insights on treatment, such as dexamethasone and tocilizumab survival of patients has since improved [32]. Whether the investigated polymorphisms alter treatment effect of these drugs is not clear. All patients in this cohort had equal access to treatment options according to the then applicable Dutch national guidelines, which included oseltamivir (as influenza occurred during this period as well) and chloroquine.

In conclusion, this study shows that carriage of the *TIRAP* polymorphism rs8177374 could be associated with lower mortality in COVID-19. This *TIRAP* polymorphism may be an important predictor in the outcome of COVID-19 and external validation is necessary to consider this association. Further research should focus on exploration of interleukin-1, interleukin-6 and tumour necrosis factor alfa levels in relation to this *TIRAP* polymorphism and their association with clinical outcomes, and whether patients with carriage of this *TIRAP* polymorphism benefit more of specific treatments targeting the TIRAP pathway.

## Acknowledgments

The authors would like to thank Ing. J. Pleijster and Ing. R. Heijmans (Laboratory of Immunogenetics, Amsterdam University Medical Center) for their technical support in the laboratory and Dr. R. de Laat-Kremers (Synapse Research Institute, Maastricht, the Netherlands) for the support with the data collection of the biochemical parameters.

## Author Contributions

**Conceptualization:** Marissa J. M. Traets, Roel H. T. Nijhuis, Servaas A. Morré, Sander Ouburg, Jasper A. Remijn, Bastiaan A. Blok, Bas de Laat, Eefje Jong, Gerarda J. M. Herder, Aernoud T. L. Fiolet, Stephan P. Verweij.

**Data curation:** Marissa J. M. Traets, Aernoud T. L. Fiolet, Stephan P. Verweij.

**Formal analysis:** Marissa J. M. Traets, Aernoud T. L. Fiolet, Stephan P. Verweij.

**Investigation:** Marissa J. M. Traets, Roel H. T. Nijhuis, Servaas A. Morré, Sander Ouburg, Jasper A. Remijn, Bastiaan A. Blok, Eefje Jong, Gerarda J. M. Herder, Aernoud T. L. Fiolet, Stephan P. Verweij.

**Methodology:** Marissa J. M. Traets, Roel H. T. Nijhuis, Servaas A. Morré, Sander Ouburg, Jasper A. Remijn, Bastiaan A. Blok, Eefje Jong, Gerarda J. M. Herder, Aernoud T. L. Fiolet, Stephan P. Verweij.

**Supervision:** Gerarda J. M. Herder, Aernoud T. L. Fiolet, Stephan P. Verweij.

**Visualization:** Marissa J. M. Traets, Aernoud T. L. Fiolet, Stephan P. Verweij.

**Writing – original draft:** Marissa J. M. Traets, Aernoud T. L. Fiolet, Stephan P. Verweij.

**Writing – review & editing:** Marissa J. M. Traets, Roel H. T. Nijhuis, Servaas A. Morré, Sander Ouburg, Jasper A. Remijn, Bastiaan A. Blok, Eefje Jong, Gerarda J. M. Herder, Aernoud T. L. Fiolet, Stephan P. Verweij.

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
