## [Decision Letter · Decision Letter 0]

31 Aug 2021

PONE-D-21-24547

Association of genetic variations in ACE2, TIRAP, and factor X with outcomes in COVID-19.

PLOS ONE

Dear Dr. Traets,

Thank you for submitting your manuscript to PLOS ONE. After careful consideration, we feel that it has merit but does not fully meet PLOS ONE’s publication criteria as it currently stands. Therefore, we invite you to submit a revised version of the manuscript that addresses the points raised during the review process.

We look forward to receiving your revised manuscript.

Kind regards,

Kanhaiya Singh, Ph.D

Academic Editor

PLOS ONE

Journal Requirements:

2. Please note that outmoded terms and potentially stigmatizing labels should be changed to more current, acceptable terminology. Examples: “Caucasian” should be changed to “white” or “of [Western] European descent”.

Additional Editor Comments:

Although the reviewers have found this study interesting, they have suggested few experiments to make this study robust.

Reviewers' comments:

Reviewer's Responses to Questions

**Comments to the Author**

1. Is the manuscript technically sound, and do the data support the conclusions?

Reviewer #1: Yes

Reviewer #2: Yes

Reviewer #3: Yes

2. Has the statistical analysis been performed appropriately and rigorously? 

Reviewer #1: Yes

Reviewer #2: Yes

Reviewer #3: Yes

3. Have the authors made all data underlying the findings in their manuscript fully available?

Reviewer #1: Yes

Reviewer #2: Yes

Reviewer #3: Yes

4. Is the manuscript presented in an intelligible fashion and written in standard English?

Reviewer #1: Yes

Reviewer #2: Yes

Reviewer #3: Yes

5. Review Comments to the Author

Reviewer #1: I suggest Author to cross validate the genotyping result with Sanger sequencing. Further It would be better to measure some biochemical parameters especially cytokines like Interleukins.

Why the author have taken plasma as a DNA source when it is possible to get intact genomic DNA from the whole blood. Plasma contains fragmented DNA and probability of having desired segment is less and thus it may hamper the genotyping result.

Reviewer #2: The research was done properly and it could be a basis for predicting clinical outcome based on the identified SNPs. It would be great if a figure can be included summarizing the genotype distribution along with the clinical outcome. For example, in Figure 3, the percentage of mortality can be included within the bar plot parts.

Reviewer #3: 1. Could the authors perform some functional validation of the TIRAP polymorphism, where population-specific TIRAP Single-Nucleotide Polymorphisms have limited impact on SARS-CoV-2 infectivity In Vitro?

6. PLOS authors have the option to publish the peer review history of their article (what does this mean?). If published, this will include your full peer review and any attached files.

Reviewer #1: No

Reviewer #2: **Yes: **Ahmed S Abouhashem

Reviewer #3: No

---

## [Author Response · Author response to Decision Letter 0]

18 Oct 2021

We would like to thank the reviewers for a careful reading of our manuscript, and for the thoughtful comments. The constructive suggestions have helped to improve the manuscript’s quality. We have copied the reviewer’s comments below in italic. Our response and changes in the manuscript follow below the comments. Moreover, we have highlighted the changes in the revised manuscript accordingly. We hope that our response and the changes in the revised manuscript will be sufficient.

Reviewer 1

I suggest Author to cross validate the genotyping result with Sanger sequencing. 

In our study real time Polymerase Chain Reactions (PCR) with specific primer pairs and probes were performed to determine the various alleles. Our PCR was successfully to determine the various alleles in 115 out of 116 patients, as explained in below. Therefore, we think there is no added value to cross validate the results with Sanger sequencing.

Further It would be better to measure some biochemical parameters especially cytokines like Interleukins.

We agree with the reviewer that measuring cytokines like interleukin-6 are of interest in patients with TIRAP polymorphism compared to patients with TIRAP wildtype. However, this study was designed to assess whether there is an association between the candidate genes (ACE2, TIRAP and Factor X) and the clinical course of SARS-CoV-2 infection. This study was not designed to determine an association between the candidate genes and biochemical parameters. The study population consists of patients who visited the emergency department with a clinical suspicion of SARS-CoV-2 infection. At that moment routinely blood was drawn of these patients. The patients presented at different days of onset of SARS-CoV-2 infection and in different clinical conditions. Therefore values of the biochemical parameters at presentation in the emergency department are expected to vary strongly. In addition, in particular levels of cytokines can increase exponentially. To detect any relevant differences, large sample sizes would be necessary. Interleukin-6 levels were not measured in our patients, so we cannot show this data. However, we did measure downstream inflammatory markers such as C-reactive protein, D-dimer and ferritin. C-reactive protein is a surrogate for the interleukin-6 pathway [1]. The results show a similar distribution between the two genotype groups at the moment of presentation at the emergency department. We added this table in the manuscript on page 12.

Table 3. Biochemical parameters in patients with TIRAP carriage and TIRAP wild-type

 TIRAP 

 Carriage (N=28) Wild-type (N=87) P-value

C-reactive protein - mg/L 108 (47-187) 85 (42-168) 0.54

D-dimer - mg/L 1.1 (0.5-2.2) 1.3 (0.6-2.2) 0.63

Ferritin - µg/L 713 (370-1752) 776 (422-1573) 0.98

Levels are depicted as median and 25th and 75th interquartiles.

We added a sentence to the statistical analysis part of the article on page 8, line number 181-182: ‘The Mann-Whitney U test was performed to compare biochemical parameters between different genotypes.’

Furthermore we emphasized the results of the table in the manuscript on the results and discussion section on the following pages:

- Page 11, line number 227-229: ‘Table 3 shows levels of C-reactive protein, D-dimer and ferritin in patients with TIRAP carriage and TIRAP wild-type. Levels of these biochemical parameters show a similar distribution between the two genotype groups at the moment of presentation at the emergency department.’ 

- Page 13-14, line number 278-288: ‘However, we did measure downstream inflammatory markers such as C-reactive protein, D-dimer and ferritin. C-reactive protein is a surrogate for the interleukin-6 pathway [27]. The results show a similar distribution between patients with TIRAP carriage and TIRAP wild-type at the moment of presentation at the emergency department. However, this study was not designed to determine an association between the candidate genes and biochemical parameters. The patients in our study presented at different days of onset of SARS-CoV-2 infection and in different clinical conditions at the emergency department. The levels of the biochemical parameters at presentation in the emergency department were expected to vary strongly and therefore cannot be interpreted or be used as functional validation of the TIRAP polymorphism. Contrary to biochemical response, the genotypes are not influenced by the day of presentation of the SARS-CoV-2 infection. Consequently, the risk of confounded effect estimates is expected to be low.’

Why the author have taken plasma as a DNA source when it is possible to get intact genomic DNA from the whole blood. Plasma contains fragmented DNA and probability of having desired segment is less and thus it may hamper the genotyping result.

We agree with the reviewer that sequencing plasma can disturb the genotyping result because plasma contains fragmented DNA. However, our desired segments of the DNA were not fragmented in the plasma of our patients. Determination of TIRAP polymorphism of one patient could not be determined, possibly due to fragmented DNA of the desired segment. We have amplified the interested segments of the DNA of 115 out of 116 patients. If there was fragmented DNA of the desired segment, it would not be possible to amplify our DNA. In conclusion, real time polymerase chain reaction with using plasma as a DNA source was successfully to determine the various alleles in 115 out of 116 patients. 

We emphasized this in the manuscript on page 11, line number 216-218: ‘Determination of the various alleles was successfully in 115 out of 116 patients. Determination of TIRAP polymorphism of one patient could not be determined due to technical issues.’

Reviewer 2

The research was done properly and it could be a basis for predicting clinical outcome based on the identified SNPs. It would be great if a figure can be included summarizing the genotype distribution along with the clinical outcome. For example, in Figure 3, the percentage of mortality can be included within the bar plot parts.

As suggested by the reviewer, we have made a figure summarizing the distribution of the outcome stratified per genotype. We agree with the reviewer that this visualization improves interpretation of the main findings of our study. No patients were carrier of Factor X polymorphism, and this was thus omitted from the figure. We added this figure in the manuscript on page 12.

 

Fig 4. Clinical outcomes stratified per genotype.

Reviewer 3

Could the authors perform some functional validation of the TIRAP polymorphism, where population-specific TIRAP Single-Nucleotide Polymorphisms have limited impact on SARS-CoV-2 infectivity In Vitro?

Our study was designed to assess whether there is an association between candidate genes (ACE2, TIRAP and Factor X) and clinical course of SARS-CoV-2 infection. Functional validation of the TIRAP polymorphism would contribute to the insights of mechanism. TIRAP has a function in the intracellular pro-inflammatory signalling pathway, and has no impact on SARS-CoV-2 receptor binding and cell entry of the virus as is known by ACE2, which is the receptor of the spike protein of SARS-CoV-2. We collected blood of patients at the moment of presentation at the emergency department. All patients in our study presented at different days of onset of SARS-CoV-2 infection and in different clinical conditions. The values of the biochemical parameters at presentation in the emergency department are expected to vary widely and therefore cannot be interpreted or be used as functional validation of the TIRAP polymorphism. Contrary to biochemical response, the genotypes are not influenced by the day of presentation of the SARS-CoV-2 infection. Consequently, the risk of confounded effect estimates is expected to be low. Future studies should collect interleukin-6 levels longitudinally during the course of a SARS-CoV-2 infection could add to understanding of the relation between carriage of the TIRAP polymorphism, different cytokine response and clinical outcomes in patients. 

References

1. Papanicolaou DA, Wilder RL, Manolagas SC, Chrousos GP. The pathophysiologic roles of interleukin-6 in human disease. Ann Intern Med. 1998 Jan 15;128: 127-37. doi: 10.7326/0003-4819-128-2-199801150-00009.

---

## [Decision Letter · Decision Letter 1]

19 Nov 2021

Association of genetic variations in ACE2, TIRAP, and factor X with outcomes in COVID-19.

PONE-D-21-24547R1

Dear Dr. Traets,

We’re pleased to inform you that your manuscript has been judged scientifically suitable for publication and will be formally accepted for publication once it meets all outstanding technical requirements.

Kind regards,

Kanhaiya Singh, Ph.D

Academic Editor

PLOS ONE

Additional Editor Comments (optional):

Reviewers' comments:

Reviewer's Responses to Questions

**Comments to the Author**

1. If the authors have adequately addressed your comments raised in a previous round of review and you feel that this manuscript is now acceptable for publication, you may indicate that here to bypass the “Comments to the Author” section, enter your conflict of interest statement in the “Confidential to Editor” section, and submit your "Accept" recommendation.

Reviewer #1: All comments have been addressed

Reviewer #3: All comments have been addressed

2. Is the manuscript technically sound, and do the data support the conclusions?

Reviewer #1: Yes

Reviewer #3: Yes

3. Has the statistical analysis been performed appropriately and rigorously? 

Reviewer #1: Yes

Reviewer #3: Yes

4. Have the authors made all data underlying the findings in their manuscript fully available?

Reviewer #1: Yes

Reviewer #3: Yes

5. Is the manuscript presented in an intelligible fashion and written in standard English?

Reviewer #1: Yes

Reviewer #3: Yes

6. Review Comments to the Author

Reviewer #1: The authors have addressed all the comments in satisfactory way. The manuscript can be accepted for the publication.

Reviewer #3: (No Response)

7. PLOS authors have the option to publish the peer review history of their article (what does this mean?). If published, this will include your full peer review and any attached files.

Reviewer #1: No

Reviewer #3: No

---

## [Editor Report · Acceptance letter]

30 Dec 2021

PONE-D-21-24547R1 

Association of genetic variations in *ACE2, TIRAP* and *factor X* with outcomes in COVID-19. 

Dear Dr. Traets:

I'm pleased to inform you that your manuscript has been deemed suitable for publication in PLOS ONE. Congratulations! Your manuscript is now with our production department. 

Kind regards, 

on behalf of

Dr. Kanhaiya Singh 

Academic Editor

PLOS ONE